# The Comparative Effect of Occupational and Musical Enrichment on Fecal Glucocorticoid Metabolite Levels in a Captive Colony of Stumptail Macaques (*Macaca arctoides*)

**DOI:** 10.3390/biology13020124

**Published:** 2024-02-17

**Authors:** Lilian Mayagoitia-Novales, Ana Lilia Cerda-Molina, María Andrea Martín-Guerrero, Emmanuel Muñoz-Zamudio, Gema R. Estudillo-Mendoza, Javier I. Borráz-León

**Affiliations:** 1Departamento de Etología, Instituto Nacional de Psiquiatría “Ramón de la Fuente Muñiz”, Ciudad de México 14370, Mexico; mayagn@imp.edu.mx (L.M.-N.); gem@imp.edu.mx (G.R.E.-M.); javier.borraz@conahcyt.mx (J.I.B.-L.); 2Licenciatura en Antropología Física, Escuela Nacional de Antropología e Historia (ENAH), Ciudad de México 14030, Mexico; maria_martin_gue@enah.edu.mx; 3Departamento de Bioterio, Instituto Nacional de Psiquiatría “Ramón de la Fuente Muñiz”, Ciudad de México 14370, Mexico; zamudio_e@imp.edu.mx; 4Investigador por México, Consejo Nacional de Humanidades, Ciencias y Tecnologías, CONAHCYT, Ciudad de México 03940, Mexico

**Keywords:** fecal glucocorticoid metabolites, enrichment programs, occupational enrichment, music enrichment, primate welfare, *Macaca arctoides*

## Abstract

**Simple Summary:**

Environmental enrichment programs (EEP) are necessary to enhance animal welfare; specially, it has been shown that occupational EEP increases behavioral diversity and reduces aggression as well as abnormal behaviors, such as stereotypies. Most primates in the wild live in social groups characterized by dominance hierarchies. However, captive populations of primates, in zoos or laboratories experience problems such as lack of visual stimulus and restricted spaces to interact. So, captivity increases the probability of alterations in normal social behaviors and tends to produce distress, thus reducing the welfare of animals. The present study shows sex, age, and rank variations in the secretion of fecal glucocorticoid metabolites (fGCM) after introducing novel objects and playing relaxing musical sounds in the enclosures of stumptail macaques (*Macaca arctoides*). After introducing novel objects, adults showed high fGCM levels compared to juveniles and subadults, indicating a higher stress response to the novel stimulus; in contrast, music and occupational enrichment decreased fGCM levels only in middle-ranking macaques, suggesting a probable relaxing effect. Overall, our results highlight the need to consider age and social differences when planning primate enrichment programs.

**Abstract:**

Environmental enrichment improves captive animal welfare by reducing stress-related behaviors. Previous studies in a captive colony of stumptail macaques (*Macaca arctoides*) reported a reduction of aggression, coprophilia, and stereotypic behaviors after an occupational enrichment program; however, the effect on stress hormones such as glucocorticoids has not been investigated yet. The goal of this study was to compare the effect of sex, age, and social rank on changes in fecal glucocorticoid metabolites (fGCM) after applying two kinds of enrichments (occupational vs. musical) in a captive colony of stumptail macaques. We collected 234 fecal samples from 25 stumptail macaques under the following conditions: (1) basal (no enrichment), (2) three weeks of occupational enrichment, and (3) three weeks of relaxing/classical music. The Generalized Estimated Equation Model showed an increase in fGCM levels after the occupational enrichment only in adult subjects (*p* = 0.003 compared to basal). The fGCM levels reached by the adults after the occupational enrichment was higher than that of juveniles (*p* = 0.002) and subadults (*p* = 0.02). Occupational and musical enrichment decreased fGCM levels only in middle-ranking individuals (*p* < 0.001 and *p* = 0.013, respectively). No sex differences were found. In conclusion, there were age and rank differences in individuals’ physiological reactivity to the effects of environmental enrichment which need to be considered when planning enrichment programs.

## 1. Introduction

Primatological research in captive colonies, such as laboratory settings or zoos, offers the opportunity of maintaining controlled social and environmental conditions. However, captivity usually reduces the time subjects spend in activities such as exploration, impoverishes sensory stimulation, and also generates abnormal behaviors (e.g., coprophagy, increased aggression, over-grooming, repetitive, and stereotyped behaviors), reducing animal welfare [1,2,3]. Environmental enrichment programs (EEP), mainly those adding complexity and novelty into the enclosures have shown improvements in captive primates’ psychological well-being by reducing the animals’ chronic stress, this approach contributes to reduce the probabilities of biased findings [1]. In addition, the literature has shown that environmental enrichments have positive effects on primate behaviors and stress physiology. For instance, physical and occupational EEP decrease the frequency of aggression and abnormal behaviors [4]; applying a cognitive task device improves cognitive performance [5], and a combination of structural and food enrichment programs reduces the levels of glucocorticoids in isolated captive animals [6].

Since listening to classical and relaxing music (e.g., rainforest, Zen, or natural sounds) improves the emotional/mental state of humans, musical enrichment of enclosures might also have a similar effect in animals [7]. However, research exploring musical enrichment in primates has been limited and mixed results have been described; for instance, it has been reported that nature sounds (unfamiliar to captive animals) do not always achieve the expected calming effects [8,9], whereas classical music has been shown to decrease aggression and abnormal behavior in zoo-housed western lowland gorillas [9]. 

Previous research in a captive colony of stumptail macaques (*Macaca arctoides*) showed that after applying an enrichment program including structural, physical, and occupational enrichment, active behaviors such as exploration increased, whereas aggression, coprophilia, and stereotypic behaviors decreased [4]. However, it is unknown whether these behavioral changes are reflected in glucocorticoid levels as a physiological indicator of stress reduction. Some literature has shown that assessing fecal glucocorticoid metabolites (fGCM) in nonhuman primates, in captivity or in the wild, provides a reliable means to determine the effect of social and environmental stressors [10,11,12]. The goal of this study was to analyze changes in fGCM levels, as a stress-relief indicator, before and after applying two contrasting kinds of enrichment programs in a captive colony of stumptail macaques: (1) occupational enrichment program (OEP), by introducing novel objects and foods, and (2) music enrichment program (MEP) by exposing the subjects to relaxing/classical sounds. Since stumptail macaques form dominant hierarchies and young individuals usually are more active and explorative than elders, we focused on analyzing whether the changes in fGCM levels after the EEP depended on the subjects’ sex, social rank and age category.

## 2. Materials and Methods

### 2.1. Subjects and Housing Conditions

The study was carried out between February and April 2018; the subjects were 25 stumptail macaques (14 females and 11 males), aged between 3 and 28 years old that are part of an outdoor captive group, composed of 26 individuals at the time of the study (we excluded a 3-month infant) (Table 1). The subjects live in three large trapezoid outdoor facilities connected to each other (6.2 m length × 1.7 m minor side × 6 m major side × 6 m height for each one) and are maintained under natural environmental conditions at the Ethology Department of the Instituto Nacional de Psiquiatría “Ramón de la Fuente Muñiz” in Mexico City, Mexico. The enclosures are provided with furniture such as hanging chains, swings, metal hoops, slides, and a running wheel (see [4] for a complete description). The facilities are cleaned daily from 07:00 to 09:00; the monkeys are fed with an early meal of fresh fruits and vegetables and a midday meal of processed monkey food (Lab Diet 5038, PMI Feeds, St. Louis, MO, USA). Clean tap water is available ad libitum.

### 2.2. Procedure of EEP, Fecal Sampling, and Determination of fGCM Levels

Over three weeks, we first collected fecal samples without any EEP (basal sampling) of the 25 subjects as follows: 8 subjects in the first week, 7 subjects in the second week, and 10 in the third week (approximately 2–4 samples × subject, avoiding sampling for approximately 24 h after a fight). Afterwards, we applied the EEP alternately for six weeks as indicated in Table 2. Feces were collected following the same basal sampling order. All the objects described in Table 2, were introduced together with the first meal in each facility from Monday to Thursday; on Friday, all the objects were completely removed, and no enrichment was applied at weekends. All materials were carefully chosen to avoid any toxic surface treatments or small holes which might trap fingers or limbs. For music stimuli, we placed one speaker per enclosure (Trust Almo 2.0 PC Speaker, 10 W Peak Power, Trust, Cleveland, OH, USA) in such a way as to ensure a medium volume inside the cages (45–55 decibels). Musical sounds were played for 1 h each session, three times a day (9:00, 11:00, 13:00). 

Fecal samples were collected 24 h after the EEP began (because of the 24 h time lag, see [12] for details in fecal sampling) between 09:00 and 13:00, and collected a total of 234 samples (2–4 weekly samples from every subject and 6–12 per individual). The collected samples were not contaminated with urine. The samples were placed in conic polypropylene tubes (15 mL) and were immediately dried in a SpeedVac^®^ Thermo Concentrator (Thermo Electron Corporation, Waltham, MA, USA) at 60 °C (approximately 8–10 h), pulverized, and frozen (−70 °C) until assayed. We determined fGCM levels by using a methanol extraction and RIA assay, following the method previously described and validated by Pineda-Galindo et al. [12] for *M. arctoides*.

### 2.3. Social Rank

We followed the method for rank calculation based on normalized David’s scores (NormDS) described in Pineda-Galindo et al. [12] and Cerda-Molina et al. [14]. Behavioral data for the calculation of NormDS of each subject (the frequency of aggressions and submissions exchanged between each pair of individuals) were collected three months prior to the EEP by using the “compete” libraries [15] for R package 0.1, computed in R 3.5.1 [16]. NormDS were calculated separately for males and females [12], and rank category was calculated based on the 50th and 75th percentiles (higher-ranking: NormDS > 75th; middle-ranking: NormDS > 50th < 75th; lower-ranking: NormDS < 50th. 

### 2.4. Data Analysis

We performed a Generalized Estimated Equation (GEE) model, suitable for dealing with autocorrelated data (i.e., repeated measures on the same subjects). We included monkey ID as the subject variable, selected the auto-regressive correlation structure which is suitable for regularly repeated measurements on the same individual, and selected the Gamma with log link function (appropriated to non-normal distribution of the fGCM levels). We introduced the levels of fGCMs as dependent variable; since age and NormDS were not correlated (r_Pearson_ = 0.13, *n* = 11, *p* = 0.69; r_Pearson_ = 0.007, *n* = 14, *p* = 0.982, males and females, respectively), we analyzed them separately. Categorical variables were introduced into the model as follows: (1) the type of EEP applied (basal condition, OEP, MEP), (2) age category (juvenile, subadult, adults), (3) rank category (higher, middle, and lower-ranking), (4) sex (male/female). Because we were interested in exploring fGCM variations after the two kinds of enrichments according to sex, age, and rank, we introduced the following interactions to the model: EEP × sex, EEP × age class, and EEP × rank category. We selected the option of pairwise comparisons for the significant interactions and selected the Bonferroni method of adjustment for multiple comparisons. We used SPSS version 23 to perform the analyses and set a *p* value ≤ 0.05 as significant. Figures represented the estimated means of the GEE model and the standard deviation of the mean.

## 3. Results

GEE revealed that the levels of fGCMs did not change according to the interaction EEP × sex (Wald X^2^ = 1.35, df = 3, 234, *p* = 0.71). However, the interactions EEP × age class, and EEP × rank were significant (Wald X^2^ = 35.75, df = 6, 234, *p* < 0.001; Wald X^2^ = 14.84, df = 6, 234, *p* = 0.02, respectively). Figure 1 indicates that compared to basal condition, the OEP significantly increased fGCM levels in adults (*p* = 0.003, Figure 1), but not in juveniles or subadults (*p* > 0.05). The levels of fGCMs in adults as a response to OEP were significantly higher than those of juveniles and subadults (*p* = 0.002 and *p* = 0.02, respectively, Figure 1). Music did not cause significant changes in fGCM levels in any age group (*p* > 0.05). 

Regarding social rank, fGCM levels decreased significantly in middle-ranking individuals after both enrichments OEP (*p* < 0.001) and MEP (*p* = 0.013) compared to basal condition (Figure 2). Neither higher nor lower-ranking individuals changed their fGCM levels after both enrichments (*p* > 0.05 in all comparisons). 

## 4. Discussion

This study provides evidence about the importance of considering that the physiological response of captive primates to the EEP varies with age and social rank. We also provided evidence that the effect of OEP was more pronounced than MEP in an outdoor captive colony of stumptail macaques. Regarding age, we found that adult macaques increased their fGCM levels after the OEP and that the reached levels were higher than those of the younger macaques. These findings suggest that novelty could be perceived as a challenging or stressing situation in adult macaques, but as an entertaining situation in the youngest. In captive colonies of *Macaca arctoides*, adults tend to manipulate and hoard objects and to drive others away [16]; although we did not record the behaviors, this reaction could initiate an aggressive conflict increasing the levels of fGCM. This is relevant because some authors have recommended to pay special attention to the kind of enrichment, since the introduction of novel objects may have undesirable effects, such as increases in competition and aggression among individuals [17]. On the contrary, it has been described that the presence of adult stumptail macaques often inhibits some behaviors in juveniles, especially when they jump over another individual [16]; hence, the occupational enrichment could distract adults reducing their oversight. Bertrand [16] described in a captive colony of stumptail macaques that play behavior in young individuals is performed in a “relaxed atmosphere”, and tense situations inhibit the behavior. The fact that no significant changes in fGCM were observed in juveniles and subadults may be explained simply because their perception of the novelty may not be stressing, instead, an opportunity for social play. In support of the stress response in the adults after the EOP, it could be that as animals age, they increase anxiety-like behaviors when exposed to stressing contexts [18]. 

Regarding rank, the fact that only middle-ranking individuals exhibited a decreased fGCM pattern after both OEP and MEP, might indicates a possible relaxing effect. Here it is important to point out that all the individuals belonging to the higher and lower-ranking places were adults; in contrast, the middle-ranking position included all juveniles and subadults, and also some adults. Thus, the reduction in fGCM levels in middle-ranking macaques could be explained in part by the influence of the playing behavior in the youngest individuals, as described above, which elicited a positive response of the enrichment program. The rank order is not necessarily correlated with age in stumptail macaques; however, since hierarchy depended on a linear aggression pattern, it is not common that juveniles hold the first places since they are not as aggressive as adults [16]. Hence, another explanation for the decreased fGCM levels in middle-ranking subjects is that their social role is more flexible and more relaxed than higher or lower-ranking positions. Previous research on *Macaca arctoides* has shown that higher-ranked adults exhibited a higher increase of fGCM levels, after an acute stressor (cage-restraining), compared to lower-ranking, who showed a delayed and dampened rise [12]; this result might indicate that a physiological stress response could differ among rank or age categories and also depend on the kind of stressors. The fact that the middle-ranking macaques maintained the reduction of fGCM levels after the music exposure, might indicate that this stimulus, similar to other species including humans, exerts a relaxing effect [8,9]. In humans for example, salivary cortisol decreases after 20 min of listening to a fast-tempo sequence of classical music [19]. One of the challenges that might stress animals living in outdoor enclosures is the exposure to unpredictable loud or aversive sounds, e.g., people and traffic noises [20]. This is the case of our captive colony of stumptail macaques since they are exposed to daily constant outside noise to which they might already be accustomed. Hence, the exposure to relaxing and classical music might have been perceived as new, but calming, especially for the young middle-ranking individuals; however, the musical stimulus was not able to exert the same effect on the other individuals. Furthermore, we did not find sex differences in fGCM levels, similar to what has been previously reported for *Macaca arctoides* [12]. 

Overall, the present findings suggest that the effects of stimulating the macaques with novel items, which encourage problem solving, exploration, tactile and auditory stimulation, depend on age and on social rank, but not on sex. In general, the effect of music on fGCM was not as clear as the OEP; more studies are still needed to test the stress or relaxing effects of different kinds of sound and music.

One important limitation of this research is that we did not collect samples long after finishing the program to observe how long the effect is preserved. Another limitation of the study is that we did not analyze possible carry-over effects of alternating both kinds of enrichments, which might contribute to explain the observed effect of music. 

## 5. Conclusions

In conclusion, we found age and rank differences in individuals’ physiological reactivity (i.e., fGCM levels) to the effects of occupational and musical enrichment in an outdoor captive colony of stumptail macaques, which need to be considered when planning future enrichment programs.

## Figures and Tables

**Figure 1 biology-13-00124-f001:**
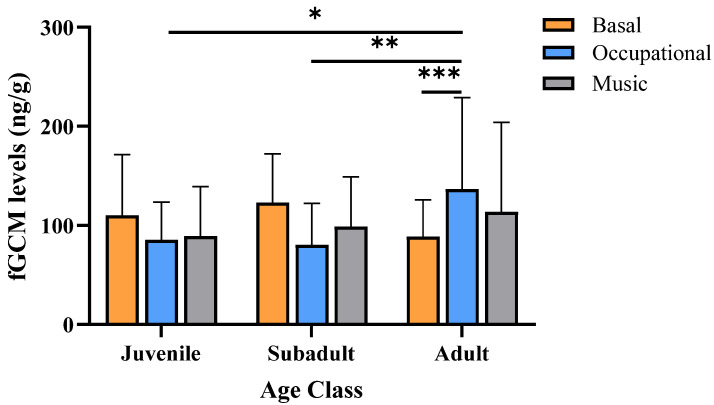
Estimated mean ± SD of fecal glucocorticoid metabolite levels (fGCM) of 25 stumptail macaques during basal (no enrichment condition) and after occupational (OEP) and musical (MEP) enrichments, according to the age classes. *** *p* = 0.003 Basal vs. OEP adults; ** *p* = 0.02 OEP adults vs. subadults; * *p* = 0.002 OEP adults vs. juveniles.

**Figure 2 biology-13-00124-f002:**
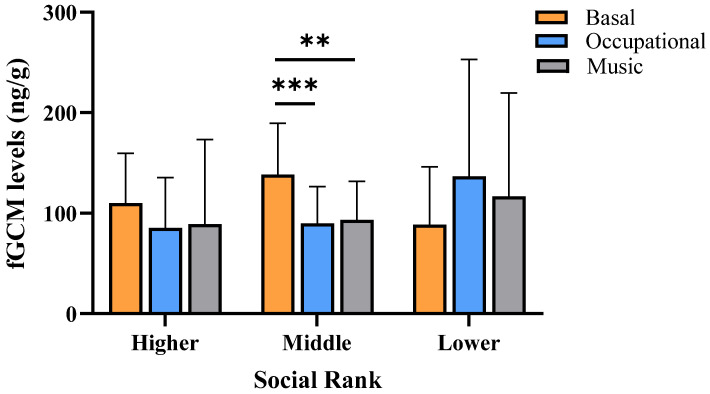
Estimated mean ± SD of fecal glucocorticoid metabolite levels (fGCM) of 25 stumptail macaques during basal (no enrichment condition) and after occupational (OEP) and musical (MEP) enrichments, according to the social rank categories. *** *p* < 0.001 Basal vs. OEP middle-ranking. ** *p* = 0.013 Basal vs. MEP middle-ranking.

**Table 1 biology-13-00124-t001:** Demographic characteristics of the 25 stumptail macaques that compose the captive colony, social rank position, and the number of fecal samples collected.

Subject	Sex	NormDS	Rank	Age (Years)	Age Category *	*n* of Fecal Samples
AU	Female	9.72	Higher	29	Adult	8
KK	Female	9.81	Higher	12	Adult	11
KT	Female	8.46	Higher	16	Adult	11
RP	Male	9.42	Higher	11	Adult	12
DF	Male	8.2	Higher	18	Adult	11
EL	Female	7.62	Middle	19	Adult	10
JU	Female	6.95	Middle	2	Juvenile	9
OLI	Female	6.89	Middle	3	Juvenile	10
LL	Female	6.81	Middle	8	Subadult	11
CH	Female	6.32	Middle	15	Adult	10
AL	Male	6.86	Middle	28	Adult	9
JI	Male	5.69	Middle	24	Adult	9
FS	Male	5.27	Middle	6	Subadult	12
FR	Male	4.67	Middle	4	Juvenile	9
OR	Male	4.64	Middle	3	Juvenile	12
RI	Female	5.84	Lower	26	Adult	8
MU	Female	5.64	Lower	21	Adult	9
CL	Female	5.36	Lower	17	Adult	8
LD	Female	5.14	Lower	20	Adult	6
AN	Female	3.78	Lower	17	Adult	11
SR	Female	3.59	Lower	15	Adult	6
PL	Male	3.68	Lower	9	Adult	9
FO	Male	2.81	Lower	11	Adult	8
RE	Male	1.92	Lower	11	Adult	6
GA	Male	1.85	Lower	24	Adult	9

NormDS = Normalized David Scores; * Age categories were based on references [12] and [13].

**Table 2 biology-13-00124-t002:** Description of the types of enrichments and fecal sampling scheduled by week in the captive colony of stumptail macaques.

Week Number	Weekday/Type of EEP	Introduced Objects/Type of Music	Sampled Subjects
1st–3rd	Mon to Fri	No enrichment (basal condition).	*n* = 25
4th	Mon and Tue Occupational	Two hanging bags made of natural fiber (around 1.5 m long) containing eight seed-filled PVC tubes and two small plastic balls each bag.	*n* = 8
	Wed and Thu Occupational	Ten small tin cans (approximately 13 cm) and four longer ones (approximately 20 cm) all with hermetic plastic covers and filled with cooked rice, sunflower seeds, and fiber cookies.	
5th	Mon to Thu Music	Classical music of Debussy (Clair de Lune, Rêverie), soft jazz (Tigran Hamasyan), Zen relaxing sounds.	*n* = 8
6th	Mon and Tue Occupational	Two wooden packaging boxes containing ten small tin cans filled with cooked rice and sunflower seeds; three longer cans filled with little pieces of carrot. Two hanging bags containing different plastic toys (plastic musical instruments, frisbees, football balloons). One car tire was introduced on each facility, surrounded by branches and leaves.	*n* = 7
	Wed and Thu Occupational	Five hanging ropes (made of natural fibers) throughout the facilities that the animals could use to swing. Twenty-five hula hoops and a tree trunk (around 30 cm diameter and 50 cm high) that the monkeys used to jump.	
7th	Mon to Thu Music	Classical music of Chopin (Nocturne op. 9 No. 2), Beethoven (Moonlight Sonata), and Zen relaxing sounds.	*n* = 7
8th	Mon and Tue Occupational	Five hanging ropes, three of them with small PVC rings which animals can slide along the ropes, similar to an abacus. Fifteen coconuts.	*n* = 10
	Wed and Thu Occupational	Twenty-five hula hoops, three plastic balloons, two car tires surrounded by branches and leaves, two tree trunks (around 30 cm diameter and 50 cm high), and twelve coconuts.	
9th	Mon to Thu Music	Zen relaxing music	*n* = 10

EEP = Environmental enrichment program. Note: on Friday we removed all the objects, and no enrichment was applied. Mon = Monday; Tue = Tuesday; Wed = Wednesday; Thu = Thursday.

## Data Availability

The data generated in this study is contained as a Appendix A.

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
