# Peer review of "The Comparative Effect of Occupational and Musical Enrichment on Fecal Glucocorticoid Metabolite Levels in a Captive Colony of Stumptail Macaques (Macaca arctoides)"

_biology, 2024, doi:10.3390/biology13020124_

Round 1
Reviewer 1 Report
Comments and Suggestions for Authors
In their paper submitted to “Biology” (2810288) Mayagoitia-Novales et al evaluated the influence of two environmental enrichment programs (EEP; occupational vs musical) on changes in faecal glucocorticoid metabolite (fGCM) levels in a captive colony of stumptail macaques. They found an influence of age and rank. I enjoyed reading the paper. The authors report a nice, small experiment, which was well performed and state-of-the-art methods applied. Below are several comments (mainly dealing with style), which I kindly ask to authors to take into account.
Specific comments as ordered by appearance in the manuscript:
Title: please add metabolites to the title, actually “fecal glucocorticoid metabolite levels” – there are no GC in the faeces, but only heir metabolites (you label this correctly in the text)
Line 26: “as a proxy of stress response” – I do not agree with this wording. You need to be careful equaling stress and GCs. You may delete this phrase, or would need to replace it with a longer description.
Line 37: Something is wrong with that sentence. I suggest: “.. to compare the effect of age and social rank on changes …”
Line 52: There are strange gaps between words here?
Line 60: Why “Then”? and better: “..enrichments reduce aggression…”
Line 61: “glucocorticoid levels” (but levels of glucocorticoids)
Line 76: I suggest using the abbreviation fGCM here and throughout! Why a small “m” – it also stands for a noun, metabolite (f is clear for the adjective fecal). Besides, please discriminate between singular/plural: fGCM levels and levels of fGCMs (throughout).
Line 80: stumptails (can you say this, or do you need to say “stumptail macaques”?) as a plural needs “form”.
2.2. and 2.3.: There is some overlap here (both have “fecal sample collection” in the heading?) Better, merge? – change “sampling” to “sample” anyway (2.2.).
How did you avoid fighting times (no sample taken ~24 h after fighting?)
Line 139: “revealed” “… did not change, neither…”
Line 146: “than those”
Figure 1: I wonder whether the concentrations were normally distributed (most likely not, because you log transformed the values; line 130). Therefore, I suggest giving boxplot graphs instead of mean and 95% CI. Those would represent the values better, and allow the reader to get a good picture of the distribution. How were multiple samples from individuals dealt with? All used, or only the mean, or? This question is also relevant for Figure 2.
What about sex differences between adults? Did you check that?
Line 166: “provides”
Line 170: delete “might”
Lines 171/172: “… younger individuals had higher basal fGCM levels than older ones…
Line 182: A thought here: all juveniles and subadults were in the middle rank class, which had the highest basal levels. May that be because they are still in the need to find their final rank position within the group?
Line 184: “.. due to their middle rank..”
Line 190: “effect… was..” Avoid the “then”.
Table 1: I wonder how the individuals of the middle rank made it into that category? A higher David Score means a higher rank, or? Why have some middle ranked individuals (FR and OR) a lower score as some lower ranked ones (MU, RI, CL and LD)?
I would love to see an additional boxplot graph showing the fGCMs of all samples of each individual.
Lines 240 and 243: Latin names should be in italics
Line 256: the year 2017 is missing.
Comments on the Quality of English Languagesee my comments above
Reviewer 2 Report
Comments and Suggestions for Authors
The comparative effect of occupational and musical enrichment on fecal glucocorticoid levels in a captive colony of stumptail macaques (Macaca arctoides)
In the above-named manuscript, the authors tested the effect of two types of enrichment (musical vs. occupational) on stress hormone levels (glucocorticoid metabolites, fGCm) in fecal samples of stumptail macaques. Test individuals (N = 25) were enriched for 6 weeks with musical and occupational enrichment alternating on a weekly basis. Both social rank and age affected the response to EEPs: young individuals had higher basal fGCm levels than adults. Juvenile and subadult fGCm levels decreased in response to EEPs while adult fGCm levels increased. In middle ranking individuals, but not in low and high ranking individuals, fGCm levels decreased in response to the EEPs. The authors conclude that age and social rank should be considered when applying EERs.
The manuscript reads well and is of interest for the readers of Biology. See below for my general and specific comments.
GENERAL COMMENTS
#1 Potential EEP carry-over effects
EEPs were alternated on a weekly basis. I wonder about carry-over effects. That is, if individuals are less stressed because they can play with novel objects: for how long are they less stressed? Are they still less stressed a couple days after the enrichment ended (i.e., when musical enrichment was applied)? Could it be that because of the alternation, there was no difference between the two enrichment methods detected?
#2 Age vs. social rank
I think describing how age and social rank are connected would add clarity to the manuscript. If these variables are intertwined, I think it would also be good to clearly state how this affects analyses and the interpretation of results.
SPECIFIC COMMENTS
Lines 20-32
The research question is not addressed, i.e., the research question is presented but not answered.
Also a few grammar mistakes, please check.
Line 43 and 169
The authors report a tendency of EEPs to decrease fGCm levels in younger individuals. But the effect is significant, correct? I would therefore not talk about a tendency (which I think means that there is a trend) but about an effect.
Line 52
Commas missing?
Line 76
Please add references that show that fGCm levels are a stress indicator.
Line 101
In Table 2, I read that EEPs were applied over 6 weeks (week 4 to week 9), not 3 weeks. Is this correct?
Line 120
How many individuals were classified as low, middle, and high ranking, respectively? Same for age – what is the N for the different age classes?
Lines 171-173
How do age and rank relate to each other? Are the youngest individuals also lower ranking?
Line 177
Where those sex differences reported in the manuscript? I didn’t see it. When discussing sex difference in the discussion I think these data should also be presented in the manuscript.
Lines 189-190
But no difference was detected between the two OEPs, right? I think this should be mentioned. I also think it is quite remarkable that 3 hours of music per day do the same for the animals in terms of enrichment as the possibility to play and explore the entire day (worth discussing?). But I wonder if it is possible to disentangle the two enrichment types (see my general comment above).
Figure 1 & 2
Are the asterisks in the right place?
Comments on the Quality of English LanguageMinor editing of English language recommended.
Reviewer 3 Report
Comments and Suggestions for Authors
I enjoyed reading the manuscript titled: “The comparative effect of occupational and musical enrichment on fecal glucocorticoid levels in a captive colony of stumptail macaques (Macaca arctoides)” The authors examined the effects of two different enrichment programs on stress levels in captive stumptail macaques. They measured stress using fecal glucocorticoid levels as a proxy. They look at differential roles of age and rank on the effects of enrichment on stress. The novel aspect of the study is in considering these differential effects.
The manuscript is generally well-written, with mostly adequate grammar and hardly any typos. There are some issues with the analyses that need to be addressed before accepting the manuscript.
I describe the issues below:
First issue is that they are interpreting main effects in an interaction model. This is inaccurate. In a model that includes interaction terms, the main effects don’t have the same interpretation as that in a model without interaction effects, especially when there are significant interactions. So, the authors should not interpret the main effect of EEP. Second, they mention that they used “Bonferroni as post hoc comparison test.” Bonferroni is not a post hoc test, rather it’s a p-value correction that is applied to the p-values from the post hoc tests to account for multiple comparisons. Due to this, it is unclear what post hoc test they have done for their comparisons, and whether they used correct tests. They need to do pairwise tests and the choice of parametric or non-parametric tests would depend on the data. Then, Bonferroni correction will need to be applied on the p-values from the tests to interpret significance. Third, from the figures, it is not clear if they have tested all pairs in each category of age (juvenile, subadults, and adults) and rank (higher, middle, lower) and between categories. There are stars on adults in Figure 1 and middle rank in Figure 2, but it is not clear which pairs they refer to. The authors should indicate which pairs the stars refer to in the figures. This may be done using horizontal lines and placing stars on the lines referring to specific pairs of interest. It is also useful to indicate which pairs are not significant, but this may be done in the text as it may crowd the figure too much.
In the abstract line 34 and on line 185 the scientific name (Macaca arctoides) should be italicized.
Round 2
Reviewer 3 Report
Comments and Suggestions for Authors
The authors have responded to my concerns adequately and the paper can be accepted.